# Neonatal Red Blood Cell Transfusion Practices: A Multi-National Survey Study

**DOI:** 10.3390/healthcare13050568

**Published:** 2025-03-06

**Authors:** Hassan Al-shehri, Ghaida Ahmad Alghamdi, Ghaida Bander Alshabanat, Bayan Hussain Hazazi, Ghadah Saad Algoraini, Raghad Abdulaziz Alarfaj, Aroob M. Alromih, Najd Mabrouk Anad Alanazi, Raghad Mabrouk Anad Alanazi, Abdullah Alzayed

**Affiliations:** 1Department of Pediatrics, College of Medicine, Imam Mohammad Ibn Saud Islamic University (IMSIU), Riyadh 13317, Saudi Arabia; aaalzayed@imamu.edu.sa; 2College of Medicine, Imam Mohammad Ibn Saud Islamic University (IMSIU), Riyadh 13317, Saudi Arabia; 442013182@sm.imamu.edu.sa (G.A.A.); 442014150@sm.imamu.edu.sa (G.B.A.); bayanhazazi@gmail.com (B.H.H.); 442013488@sm.imamu.edu.sa (G.S.A.); 442013954@sm.imamu.edu.sa (R.A.A.); aroobalromih@gmail.com (A.M.A.); nmaalanazi62@sm.imamu.edu.sa (N.M.A.A.); rmaalanazi99@sm.imamu.edu.sa (R.M.A.A.)

**Keywords:** blood transfusion, multi-country, neonates

## Abstract

**Background:** Blood transfusion is a highly critical life-saving factor in neonates, especially in extremely low birth weight infants. There is a significant lack of consensus on optimal blood transfusion methods for neonates. **Aim:** To investigate and analyze blood transfusion practice in neonates among neonatologists and neonatal nurses in a multi-country pattern. **Methods:** From September 2023 to June 2024, a cross-sectional questionnaire-based study was conducted to collect data on global blood transfusion practices in neonates. A questionnaire, developed through an extensive literature review, was distributed to neonatologists and neonatal nurses primarily via e-mail, with additional distribution via social media platforms. **Results:** This study included a total of 180 neonatologists and neonatal nurses from 27 different countries. Almost 37.7% were working in a level 3 NICU. Approximately 37.7% of the participants stated that they transfuse blood within three hours, and approximately 45.5% stated they usually use 15 mL/kg of blood. After receiving a transfusion, 99.4% of the participants mentioned that they continue to check the vital signs. More than half (72.2%) of NICU practitioners use filters when giving blood. Regarding written instructions and guidelines in the unit for blood transfusion, the majority (84.4%) stated having them in their units, of which, 86.8% mentioned that blood transfusion threshold stated in the guidelines either using hemoglobin or hematocrit. **Conclusions:** This study found variability in blood transfusion practices around the world. While most have developed neonatal blood transfusion guidelines, certain countries still lack national protocols. Establishing comprehensive guidelines is essential to standardizing procedures, thereby minimizing the risk of inappropriate or unsafe blood transfusions in this neonatology practice.

## 1. Background

Blood transfusion is a highly critical life-saving factor in neonates, especially in extremely low birth weight (ELBW) infants. About 90% of ELBW infants with a birth weight of less than 1000 g will receive at least one red blood cell transfusion during their initial hospital stay [1,2,3]. In addition, 60% of very low birth weight (VLBW) infants will receive RBC transfusions during their neonatal intensive care unit (NICU) stay [4]. Neonates face higher transfusion risks than other age groups, including metabolic complications like hypoglycemia, hyperkalemia, and hypocalcemia. They can also experience immunologic issues such as hemolytic reactions, allergic reactions, and transfusion-related acute lung injury (TRALI). Blood transfusions may transmit infectious agents like viruses and bacteria. Transfusion therapy requires special considerations [4]. Blood transfusion practice typically follows guidelines regarding preparation, indication, storage, and administration to ensure safety [5,6]. Nevertheless, an international survey showed that only half of NICUs worldwide use protocols to guide their local practices [7]. At the same time, other countries do not have such clear national guidelines [5,8]. These variations in transfusion practices among NICUs worldwide reveal a significant lack of consensus on optimal blood transfusion methods for neonates.

Transfusion decisions about RBCs have to be based upon the balance between benefit and harm [9]. RBC transfusions may improve cerebral oxygenation, which in turn has been linked to a reduced risk of death or neurodevelopmental impairment [10]. On the other hand, RBC transfusion can lead to increased risks of serious side effects. For this reason, the full benefit–harm relationship for RBC transfusion in very preterm neonates needs to be fully realized in order to guide clinical decision-making. To address these inconsistencies and improve care standards, this study aimed to investigate and analyze blood transfusion practices among neonatologists and neonatal nurses across multiple countries. By examining these practices, the study aims to identify similarities and differences, providing insights that could guide the development of more standardized and effective transfusion protocols for neonatal care. The study aimed to include neonatologists and neonatal nurses from different countries, specifically focusing on nations in the Middle East, Europe, Canada, the USA, and India.

## 2. Methods

### 2.1. Study Design

This is an online cross-sectional survey study that was conducted between September 2023 and June 2024 to examine current neonatal blood transfusion practices.

#### 2.1.1. Sample Population and Recruitment

This study employed the convenience sampling technique to invite participants who meet the inclusion criteria to participate in the study. It is deemed as an appropriate, time-saving, and effective procedure to invite participants who are accessible and welling to participate.

Neonatologists and neonatal nurses who are currently practicing their profession formed the study population for this study and were surveyed to investigate their practices on neonatal blood transfusion. We invited our targeted study population to participate in the study through social media platforms (WhatsApp, X app, and Telegram) and through e-mails. In addition, the search for eligible participants’ authenticated contacts (emails or phone numbers) was conducted though universities websites, hospital websites, and published articles. Available contact information was utilized to contact them and invite them to participate in the study. The specific objectives of the study were presented in the cover letter of the questionnaire which served as an invitation letter for response. Invitations to participate were extended to only those who a priori satisfied the eligibility criteria. Inclusion criteria were listed in the cover letter to the respondents. Participants were assured that their data would be treated with utmost confidentiality. An informed consent statement was prominently displayed on the first page of the survey, ensuring that participants were fully informed and consented to participate in the study.

#### 2.1.2. The Study Tool

This study used an adapted version of a previously developed questionnaire that examined neonatal platelet transfusion practices [11]. The questionnaire tool comprised of 12 questions in a multiple choices format (refer to Appendix A). There were two sections in this questionnaire-based study. The first section was about the participants’ practice characteristics. In the first section, participants were asked about their practice characteristics, including country of practice, years of experience, level of NICU, and availability of written guidelines for neonatal blood transfusions (including thresholds). Furthermore, they were questioned about which vascular access they used to transfuse red blood cells and whether they used a filter during the transfusion. The second section was about blood transfusion triggers, which included questions that addressed blood transfusion triggers in two common NICU clinical scenarios.

#### 2.1.3. Questionnaire Piloting

Expert neonatologists examined the proposed questionnaire to ensure it met external validity standards. A second evaluation was carried out by skilled neonatologists. After the assessment program, a group of neonatologists participated in a pilot trial. The pilot study results demonstrated that the questions accurately represented the groups’ real practices and were easy to grasp.

#### 2.1.4. Statistical Analysis

The Statistical Package for Social Science software (version 29) (IBM Corp, Armonk, NY, USA) was used to analyze the data for this study. Categorical variables were presented as frequencies and percentages. The Chi-squared test was used to examine the difference in blood transfusion practices between participants who had written guidelines for red blood cell transfusion in their unit and those who did not. A *p*-value < 0.05 was considered statistically significant.

## 3. Results

### 3.1. Participants’ Country of Practice

This study included a total of 180 neonatologists and neonatal nurses from 27 different countries. Around one-quarter of the study participants (24.4%) were from the United States; Table 1 presents the distribution of country of practice for the study participants.

#### 3.1.1. Participants’ Practice Characteristics

Around one-third of the study participants (33.9%) reported that they have 1–5 years of experience in the NICU. Almost 37.7% were working in a level 3 NICU. A total of 1.1% of the participants reported that their duration of the pRBC transfusion is 30 min, 3.8% reported 1 h, 21.6% reported 2 h. In addition, 37.7% of the participants stated that they transfuse blood for three hours, and approximately 45.5% stated they usually use 15 mL/kg of blood. After receiving a transfusion, 99.4% of the participants mentioned that they continue to check the vital signs. More than half (72.2%) of NICU practitioners use filters when giving blood. Regarding written instructions and guidelines in the unit for blood transfusion, the majority (84.4%) stated having them in their units, of which 86.8% mentioned that the blood transfusion threshold was stated in the guidelines either using hemoglobin or hematocrit. There was a statistically significant difference in the volume of blood typically ordered for neonatal transfusions and the use of filters for red blood cell administration between units that had written guidelines for red blood cell transfusion and those that did not (*p* < 0.05), Table 2.

#### 3.1.2. Blood Transfusion Triggers

Table 3, below, presents two common NICU clinical scenarios that involve blood transfusion. In scenario 1, a three-day-old preterm infant born at 24 weeks of gestation was intubated and on maximum ventilatory support with a Fio2 of 70%. A hemoglobin threshold ≤ 10 gm/dL was the most common response (41.1%) for red blood cell transfusion in this scenario. Comparing those who had written guidelines to those who did not, it was seen that 35.5% of the participants who had written guidelines selected ≤ 10 gm/dL, while among those without written guidelines, 70.5% selected ≤ 10 gm/dL. There is a statistically significant difference in the responses of the two groups (*p*-value: 0.005). In scenario 2, a three-day-old preterm infant, born at 24 weeks of gestation, is stable on minimal ventilation and requires a Fio2 of 21%. Overall, for this scenario, 58.3% of participants chose ≤ 10 gm/dL as their transfusion threshold. Of the participants who reported having written guidelines, 56.6% chose this threshold, whereas of those participants that did not report having guidelines, 67.9% chose this threshold. For this scenario, there is no statistical difference between groups (*p*-value: 0.807).

## 4. Discussion 

Blood transfusion is crucial for saving the lives of neonates, particularly those with extremely low birth weights. Rather than solely addressing iatrogenic losses, transfusions are typically performed to correct low hematocrit levels and alleviate clinical symptoms and signs [12]. The blood volume transfused to a newborn is critical, especially in extremely premature newborns. A small amount may necessitate more transfusions, whereas a large volume may cause cardiopulmonary damage due to volume overload. In addition, high transfusion volumes slow erythropoiesis [13].

Clinical practice guidelines for red blood transfusion in very preterm neonates recommend a restrictive RBC transfusion strategy based on hemoglobin concentration [10]. The suggested hemoglobin thresholds changed depending on the postnatal week and the necessity for respiratory support [10]. Because hemoglobin may be a more relevant measure (although a surrogate sign) of blood oxygenation capacity, hemoglobin concentration was chosen as the preferable laboratory test for transfusion threshold [14].

A recent study from 64 NICUs in 22 European countries examined RBC transfusion practices in NICUs between 2022 and 2023 [15]. This study reported that 82.8% of transfusions were given based on a defined Hb threshold. Hemoglobin levels before transfusions indicated thresholds below the restrictive thresholds set by Transfusion of Prematures (TOP) randomized controlled trial in 36.4% and Effects of Transfusion Thresholds on Neurocognitive Outcomes of Extremely Low-Birth Weight Infants (ETTNO) randomized controlled trial in 44.4% of transfusions [15]. On the other hand, they were above liberal thresholds in 7.3% and 7.5% of transfusions, respectively, and between restrictive and liberal thresholds in 48.3% and 56.1% of transfusions, respectively. Moreover, 63.7% of transfusions given based on a threshold had volumes of 15 mL/kg and were administered over 3 h for 54.2% of the cases; however, there was substantial variation in dose and duration [15].

In this study, the most typical transfused red blood cell volume was 15 mL/kg. Some units take into account additional factors, such as the neonate’s hemoglobin level or weight. Transfused RBCs most frequently ranged from 10 to 20 mL/kg, according to the transfusion practices in Switzerland [5]. The survey provides valuable insights into the current neonatal blood transfusion practices across multiple countries. Our study included a diverse sample of 180 neonatologists and neonatal nurses, with a majority (33.9%) having 1–5 years of job experience and about 37.7% working in level 3 NICUs, indicating representation from high-acuity neonatal care settings. The survey revealed several notable findings regarding transfusion practices. Approximately 37.7% of respondents reported transfusing blood within 3 h, and 45.5% typically used 15 mL/kg of blood, suggesting a general alignment with recommended transfusion volumes and durations. Furthermore, a vast majority (99.4%) continued to monitor the neonate’s vital signs during and after the transfusion, and the majority (72.2%) used filters when administering blood transfusions, highlighting the importance placed on patient safety and the mitigation of potential complications. Most participants (84.4%) stated that their NICU had written instructions and guidelines for blood transfusion, indicating an effort to standardize practices, though the remaining 15.5% without formal guidelines suggests a need for further development and implementation of evidence-based protocols. Additionally, our survey revealed that 30% of neonatologists use a peripheral line to transfuse RBCs. However, some neonatologists prefer using an umbilical venous catheter or a peripherally inserted central catheter (PICC) line. A dedicated peripheral line is considered the optimal intravenous access method, as it allows for slow transfusion rates and helps prevent infection. Nonetheless, inserting a peripheral intravenous cannula in extremely premature babies or those in poor medical conditions can be challenging [5,16,17]. UVC is frequently used in the NICU, as in high-risk newborns it provides safe vascular access immediately after birth [18]. UVCs are typically used for blood transfusions and IV administration of parenteral nutrition and medications [18]. Moreover, the use of UVC in term neonates is influenced by age of the neonate, the infusate characteristics, and expected duration of therapy [19].

Usually, blood issued for neonatal transfusion is group O packed RBCs with compatible infant Rh type. However, if passive maternal anti-A or anti-B is not detected, non-group O infants can be issued with non-group O RBCs [20]. During a perinatal emergency, it is necessary that all birthing centers should have supporting transfusion services prepared to issue uncross-matched group O Rh-negative blood [9]. It is estimated that the hematocrit of packed RBCs with buffy coat preparation is around 60% [9].

This survey also explored transfusion triggers by presenting two clinical scenarios. For a preterm infant delivered at 24 weeks gestation with respiratory distress, the majority (41.1%) selected a hemoglobin threshold of ≤10 gm/dL for transfusion. Similarly, for a stable 24-week preterm infant on minimal ventilatory support, around 58.3% of participants selected a hemoglobin threshold of ≤10 gm/dL for transfusion. Based on the recent clinical guidelines published for RBC transfusion thresholds in very preterm neonates, the recommended hemoglobin thresholds should vary based on respiratory support needs and postnatal week [10]. At the first, second, and third postnatal weeks or more, for neonates on respiratory support, the recommended thresholds should be 11, 10, and 9 gm/dL, respectively. In addition, for neonates on no or minimal respiratory support, the recommended thresholds should be 10, 8.5, and 7 gm/dL, respectively [10].

In clinical practice, RBC transfusions for infants are delivered using an infusion pump and provided in bags or syringes. Rewarming is the first important step to be followed, with a maximum duration of 4 h imposed on each issue. Conventional transfusion volumes range from 10 mL/kg to 20 mL/kg; however, larger amounts may elevate RBC volume by 50% and induce significant variations in hemoglobin levels and viscosity [20]. Newborns can tolerate transfusion volumes of 20 mL/kg administered at a rate of 7 mL/kg/h, although the assessment of this practice is limited; a cautious maximum rate would be 5 mL/kg/h [21]. An alternate method involves calculating the volume of blood transfused based on hemoglobin target selection, traditionally set at 15 gm/dL during the early neonatal period. Striving to achieve this objective later in life (when the transfusion threshold has decreased to 7.5 gm/dL) leads to excessively large transfusion volumes; hence, a lesser target (e.g., 13 gm/dL) may be used. Smaller and more frequent blood transfusions have gained greater acceptance as the hazards associated with donor exposure have diminished [20]. Due to the existing ambiguity concerning the neurodevelopmental implications of anemia, it may be advisable to maintain stable hemoglobin levels [20].

The survey findings highlight several key points. The study population was made up of practitioners from many different countries and NICUs. This shows how neonatal blood transfusions are performed around the world and may help explain the differences seen in transfusion protocols, thresholds, and administration methods. The majority of respondents reporting the use of written guidelines for blood transfusion indicates an effort to standardize practices, though the remaining 15.5% without formal guidelines suggests a need for further development and implementation of evidence-based protocols. However, according to the international survey of transfusion practices for extremely premature infants, half of the respondents (51.1%) reported having a written policy with specific red cell transfusion guidelines in their unit [7]. The consistency in selecting similar hemoglobin thresholds for transfusion is an encouraging finding, as it demonstrates an evidence-based approach to transfusion decision-making. Overall, these findings can inform future efforts to standardize and optimize neonatal blood transfusion protocols globally, ultimately, contributing to improved patient outcomes in neonatal intensive care.

This study faced some limitations; the number of participants varied from each country, there was a high representation of participants from the USA (24.4%), and the number of participants from other countries was deficient, which might affect the generalizability of our study findings. Furthermore, the cross-sectional design and use of a self-administered online survey may reflect response bias. Therefore, these findings should be interpreted carefully.

## 5. Conclusions

According to the study, most NICUs worldwide have written neonatal blood transfusion guidelines. Nonetheless, there are some countries that lack unit guidelines. Furthermore, research revealed that most participating neonatologists transfuse red blood cells using peripheral lines. As blood transfusions are common practice among neonatologists, it is necessary to provide guidelines for the transfusion in order to standardize procedures and reduce the use of potentially unsuitable and unsafe blood transfusion practices.

## Figures and Tables

**Table 1 healthcare-13-00568-t001:** Country of practice.

Variable	Frequency	Percentage
United States	44	24.4%
Oman	31	17.2%
Egypt	27	15.0%
Saudi Arabia	21	11.6%
United Kingdom	11	6.1%
India	10	5.5%
Canada	5	2.7%
Bahrain	4	2.2%
Australia	3	1.6%
Brazil	3	1.6%
Italy	3	1.6%
United Arab Emirates	3	1.6%
Argentina	1	0.5%
Austria	1	0.5%
Ireland	1	0.5%
Jordan	1	0.5%
Korea	1	0.5%
Kuwait	1	0.5%
Mexico	1	0.5%
Nepal	1	0.5%
Netherlands	1	0.5%
Pakistan	1	0.5%
Peru	1	0.5%
Philippines	1	0.5%
Qatar	1	0.5%
Spain	1	0.5%
Tunisia	1	0.5%

**Table 2 healthcare-13-00568-t002:** Participants’ practice characteristics.

	Overall		Have a Written Guideline for Blood Transfusion		Do Not Have a Written Guideline for Blood Transfusion		
Variable	Frequency	Percentage	Frequency	Percentage	Frequency	Percentage	*p* Value
Years of experience in NICU:							
1–5 years	61	33.9%	51	33.6%	10	35.7%	0.308
6–10 years	46	25.5%	38	25%	8	28.6%	
11–20 years	46	25.5%	37	24.3%	9	32.1%	
Over 20 years	27	15%	26	17.1%	1	3.6%	
NICU practicing settings:							
Level 1 NICU	33	18.3%	24	15.8%	9	32.2%	0.134
Level 2 NICU	19	10.5%	15	9.9%	4	14.3%	
Level 3 NICU	68	37.7%	59	38.8%	9	32.2%	
Level 4 NICU	60	33.3%	54	35.5%	6	21.4%	
Duration of the PRBC transfusion: §							
30 min	2	1.1%	2	1.3%	0	0%	0.539
1 h	7	3.8%	6	3.9%	1	3.6%	
2 h	39	21.6%	30	19.7%	9	32.1%	
3 h	68	37.7%	57	37.5%	11	39.3%	
4 h	64	35.5%	57	37.5%	7	25%	
How much do you typically order for neonatal blood transfusion?							
10 mL/kg	80	44.4%	62	40.8%	18	64.3%	0.031 *
15 mL/kg	82	45.5%	72	47.4%	10	35.7%	
>15 mL/kg	18	10%	18	11.8%	0	0%	
Do you monitor the neonate’s vital signs during blood transfusions even though they have not been monitored before?							
No	1	0.5%	1	0.7%	0	0%	1.00
Yes	179	99.4%	151	99.3%	28	100%	
Which vascular access do you use to transfuse red blood cells? §§							
Single-line access							
Peripheral line	54	30%	46	30.3%	8	28.6%	0.111
PICC line	8	4.4%	4	2.6%	4	14.3%	
UVC	1	0.5%	1	0.7%	0	0%	
Dual-line access							
Peripheral line, PICC line	9	5%	7	4.6%	2	7.1%	
Peripheral line, UVC	66	36.6%	55	36.2%	11	39.3%	
UVC, PICC line	6	3.3%	6	3.9%	0	0%	
Triple-line access							
Peripheral line, UVC, PICC line	36	20%	33	21.7%	3	10.7%	
Do you use a filter for the administration of red blood cells?							
No	50	27.7%	37	24.3%	13	46.4%	0.030 *
Yes	130	72.2%	115	75.7%	15	53.6%	
Do you have written guidelines for neonatal blood transfusion in your unit?							
No	28	15.5%					
Yes	152	84.4%					
If you answered (Yes) in the previous question, was the blood transfusion threshold stated in the guidelines either using hemoglobin or hematocrit? (n = 152)							
No	20	13.2%					
Yes	132	86.8%					

NICU = Neonatal Intensive Care Unit; PRBC = Packed Red Blood Cells; PICC = Peripherally Inserted Central Catheter; UVC = Umbilical Vein Catheter. § Percentages reported represent distinct percentages rather than cumulative percentages; §§ Single answer question. * *p* < 0.05

**Table 3 healthcare-13-00568-t003:** Blood transfusion triggers.

Overall			Have a Written Guideline for Blood Transfusion		Do Not Have a Written Guideline for Blood Transfusion		*p* Value
Variable	Frequency	Percentage	Frequency	Percentage	Frequency	Percentage	
A three-day-old preterm born at 24 weeks of gestation, intubated on maximum ventilatory support, Fio2 70%							
≤10 gm/dL	74	41.1%	54	35.5%	20	70.5%	0.005
≤12 gm/dL	57	31.7%	55	36.2%	2	7.1%	
≤15 gm/dL	28	15.6%	25	16.4%	3	10.7%	
No specific threshold	21	11.7%	18	11.9%	3	10.7%	
A three-day-old preterm born at 24 weeks, stable on minimal ventilation setting, Fio2 21%							
≤10 gm/dL	105	58.3%	86	56.6%	19	67.9%	0.807
≤12 gm/dL	37	20.6%	32	21.1%	5	17.9%	
≤15 gm/dL	15	8.3%	14	9.2%	1	3.6%	
No specific threshold	23	12.8%	20	13.2%	3	10.7%	

## Data Availability

The datasets used and/or analyzed during the current study are available from the corresponding author on reasonable request.

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
