# Peer review of "Neonatal Red Blood Cell Transfusion Practices: A Multi-National Survey Study"

_healthcare, 2025, doi:10.3390/healthcare13050568_

Round 1

Reviewer 1 Report

Comments and Suggestions for Authors

Author Response

Reviewer 1:

In this study, Al-shehri et al. describe the results of an online international survey completed by 180 neonatologists and neonatal nurses in 27 unique countries. Overall, the survey provides valuable insight into blood transfusion practices in a variety of settings. There are several areas that require revision or clarification.

Background

  1. The citation for the following statement is missing. ‘An international survey showed that only half of NICUs worldwide use protocols to guide their local practices.’ Moreover, how does the current manuscript/survey differ from the prior international survey?

- Thank you for this comment. We have now added the missed citation to support this statement, see line 47. Our study provides an updated findings concerning neonatal blood transfusion practices from multiple countries in the Middle east region and other countries including the United States, the United Kingdom, India, Canada, Australia, Brazil, and others. This will give more comprehensive insights concerning these practices.

Methods

  1. The authors mention that they study amin was specifically focusing on nations in Middle East, Europe, Canda, the USA, and India. Why were those specific countries the focus? How did the authors target providers in specific countries via the online survey?

- Thank you for this comment. As mentioned above, examining blood transfusion practices from different countries in the Middle east region and other countries including the United States, the United Kingdom, India, Canada, Australia, Brazil, and others provides better and more comprehensive understanding for these important practices and reflect global variation. Besides, this could reflect the impact of applying international guidelines versus clinical judgment (where some countries do not have specific strict guidelines). Concerning survey link dissemination, this was done through searching for eligible participants authenticated contacts (emails or phone numbers) in universities websites, hospital websites, and published articles. Available contacts information was utilized to contact them and invite them to participate in the study. Besides, social media platforms were utilized. We have now highlighted these details in the method section, see lines 77-80.

  1. It would be helpful to the readers if the 12-question questionnaire was included as supplemental material.

- Thank you for this comment. We have now added the questionnaire tool as supplemental material.

  1. The questionnaire was evaluated first by expert neonatologists and then by skilled neonatologists. What is the objective difference between these two groups?

- Thank you for this comment. Expert neonatologists involved in the piloting phase were chosen as they are more experienced in both clinical practice and research (so they can evaluate the tool from scientific and researchers’ point of view), while skilled neonatologists were selected based on their deep clinical experience and practices in patients care (so they can evaluate the questionnaire from healthcare practitioners’ point of view in terms of its understandability).  

Results

  1. There were 180 neonatologists and neonatal nurses that participated in the survey. It would be helpful to include the frequency/percentage of the respondents that are neonatologists versus neonatal nurses.

- Thank you for this comment. Unfortunately, we did not aim to examine the difference between neonatologists and neonatal nurses in this study, therefore, we did not ask for this information.

  1. Canada was one of the previously mentioned countries of specific interest yet respondents from Canada only made up 2.7% of the respondents.

- Thank you for this comment. As mentioned above, examining blood transfusion practices from different countries in the Middle east region and other countries was the aim of this research. We contacted potential participants from multiple countries including Canada, however, limited number of participants responded to our survey and participated in the study from Canada.

  1. Regarding duration of the pRBC transfusion, it is stated 37.7% of respondents transfuse blood within three hours, yet the table also reports that 1.1% transfused in a duration of 30 minutes, 3.8% transfused in a duration of 1 hour, and 21.3% transfused in a duration of 2 hours. Depending on the terminology used in the question and answers, the results are misleading because it seems like 64.2% of the respondents transfused within 3 hours. Please clarify and correct as needed.

- Thank you for this comment. The percentages mentioned in Table 2 presents distinct categories rather than cumulative one. Therefore, 1.1% of the participants reported that their duration of the pRBC transfusion is 30 minutes, 3.8% for 1 hour, 21.6% for 2 hours, and 37.7% for three hours. We have now clarified this in the footnote of the table and in the text, see lines 120-122.

  1. Regarding the results from the clinical scenario questions, the authors state ‘a transfusion threshold ≤ 10 gm/dl was the most common response (41.1%) for red blood cell transfusion in this scenario.’ I presume this threshold is referring to hemoglobin level but that should be specifically stated.

- Thank you for this comment. Yes, we have now corrected this, see line 138.

  1. The authors state ‘Again, for this scenario, there is no statistical difference between groups (P- value: 0.807).’ Please remove the word again as this implies a similar result as the first scenario which is not the case, as the first scenario had a statistically significant difference (P-value:

0.005).

- Thank you for this comment. We have now addressed this point, see line 148.

  1. The table results for vascular access appear confusing. Please clarify what the different responses mean.

- Thank you for this comment, we have now organized these findings in Table 2 grouping them based on the number of access types.

 Discussion

  1. Please provide a citation for this statement. ‘In addition, high transfusion volumes slow erythropoiesis.’

- Thank you for this comment, we have now added the citation, see line 159.

  1. ‘This survey also explored transfusion triggers by presenting two clinical scenarios. For a preterm infant delivered at 24 weeks gestation with respiratory distress, the majority (41.1%) selected a hemoglobin threshold of ≤ 10 gm/dl for transfusion. Similarly, for a stable 24-week preterm infant on minimal ventilatory support, around 58.3% of participants selected a hemoglobin threshold of ≤ 10 gm/dl for transfusion. The fact that the survey participants consistently chose similar hemoglobin thresholds for transfusion, even in different clinical situations, suggests that they generally followed current recommendations and guidelines. This shows that they made transfusion decisions based on evidence.’. I disagree with the authors’ conclusions of these findings since the recent clinical guidelines published in JAMA 2024 for preterm neonates in the first week of life should be different for the two clinical situations based on need for respiratory support (11 g/dL vs 10 g/dL). This would suggest the choice of <10 g/dL is the incorrect choice for the first scenario based on guidelines. Of note, 36.2% of those with written guidelines chose < 12 gm/dl and 7.1% of those without written guidelines chose < 12 gm/dl. This may be worthwhile discussing.

- Thank you for this comment, we have now addressed this point and added the recommendations from Deschmann et al. study in JAMA based on the reviewer’s comment, see lines 225-231.

  1. The authors use the units gm/dl earlier in the manuscript, though in the sixth paragraph of the discussion units g/L are utilized. This is confusing for the reader.

- Thank you for this comment, we have now addressed this comment in the sixth paragraph in the discussion.

Conclusions

  1. The conclusions state ‘according to the study, most neonatologists worldwide have written neonatal blood transfusion guidelines’ yet the study was sent to both neonatologists and nurses. Recommend changing the statement to ‘most NICUs worldwide…’

- Thank you for this comment, we have now addressed this comment, see line 271.

  1. The conclusions also state ‘there are some countries that lack national guidelines.’ It is important to note that having unit guidelines and having national guidelines are not always the same thing. Wording should be adjusted based on the actual survey question/answer.

- Thank you for this comment, we have now addressed this comment, see line 272.

Reviewer 2 Report

Comments and Suggestions for Authors

Dear colleagues,

I was honored to review a well-structured and engaging paper on blood transfusion practices in neonates worldwide. Congratulations.

However, I have some minor comments:

  1. Perhaps the title should be completed to better reflect the content, and instead of ”blood,” red blood cell” would be more appropriate.
  2. The Abstract is well structured, reflecting the basics of the chosen methodology, the most important results of the study, and conclusions per the results.
  3. The Introduction section is well presented.
  4. In the Material and Methods section, could the authors explain more about how the respondents were chosen from the social media platforms? How ”convenience sampling” was performed?
  5. From Table 2, on the question on vascular access, I suggest stating if the authors were asking for a single answer (as it seems by adding the percentages of the answers) or multiple choices. If the questions admitted a single answer, this could change the discussions on the results as the umbilical lines are available only at birth and are not accessible for later transfusions.
  6. Citations can be offered for the statement in lines 170 (the duration of the transfusions), 180-181 (the vascular access choice), and 194-196 (the transfusion threshold).
  7. The Results section is well structured, although most of the results are also commented on in the Discussion chapter.
  8. In the Discussion chapter, comparative comments could also be offered regarding the transfusional practices as presented in recent papers (for example, in Europe; I suggested some references in the analysis of the paper's references)

Author Response

Reviewer 2:

I was honored to review a well-structured and engaging paper on blood transfusion practices in neonates worldwide. Congratulations.

However, I have some minor comments:

1. Perhaps the title should be completed to better reflect the content, and instead of ”blood,” red blood cell” would be more appropriate.

- Thank you for this comment, we have now addressed this comment, see line 1.

2. The Abstract is well structured, reflecting the basics of the chosen methodology, the most important results of the study, and conclusions per the results.

- Thank you for this comment.

3. The Introduction section is well presented.

- Thank you for this comment.

4. In the Material and Methods section, could the authors explain more about how the respondents were chosen from the social media platforms? How ”convenience sampling” was performed?

- Thank you for this comment. We have now added further details concerning the recruitment procedure, see lines 77-80.

5. From Table 2, on the question on vascular access, I suggest stating if the authors were asking for a single answer (as it seems by adding the percentages of the answers) or multiple choices. If the questions admitted a single answer, this could change the discussions on the results as the umbilical lines are available only at birth and are not accessible for later transfusions.

- Thank you for this comment. We have now highlighted this point in the discussion, see lines 202-207.

6. Citations can be offered for the statement in lines 170 (the duration of the transfusions), 180-181 (the vascular access choice), and 194-196 (the transfusion threshold).

- Thank you for this comment. We have now clarified that these are our study findings and do not require citations.

7. The Results section is well structured, although most of the results are also commented on in the Discussion chapter.

- Thank you for this comment.

8. In the Discussion chapter, comparative comments could also be offered regarding the transfusional practices as presented in recent papers (for example, in Europe; I suggested some references in the analysis of the paper's references)

-Thank you for this comment. We have now added transfusional practices as presented in recent paper from 64 NICUs in 22 European countries, see lines 166-177.